# Disentangling the Effects of Mindfulness, Savoring, and Depressive Symptoms among Emerging Adults

**DOI:** 10.3390/ijerph20085568

**Published:** 2023-04-18

**Authors:** Rebecca Y. M. Cheung, Melody C. Y. Ng

**Affiliations:** 1School of Psychology and Clinical Language Sciences, University of Reading, Reading RG6 6ES, UK; 2Centre for Child and Family Science, The Education University of Hong Kong, Hong Kong SAR, China; 3Department of Psychology, The Chinese University of Hong Kong, Hong Kong SAR, China

**Keywords:** savoring, mindfulness, depressive symptoms, emerging adults, positive experiences

## Abstract

The links between mindfulness, savoring positive experiences, and depressive symptoms have been well established in the literature. Nevertheless, little has been done to disentangle the prospective relations among these constructs. Clarifying the longitudinal relations is crucial, as it enables researchers and practitioners to discern potential trajectories of mental health interventions. In this study, 180 emerging adults between 18 and 27 years old were recruited twice at 3 months apart to complete self-reported measures including mindfulness, savoring positive experiences, and depressive symptoms. Cross-lagged path analysis indicated that savoring the moment was predictive of mindfulness three months later, whereas depressive symptoms were predictive of both mindfulness and savoring the moment three months later, above and beyond the effects of age, gender, and family income. Additionally, mindfulness, savoring positive experiences, and depressive symptoms were significantly correlated at baseline. The present study evidenced short-term inverse effects of depressive symptoms on mindfulness and savoring the moment, as well as a positive effect of savoring the moment on mindfulness. Hence, interventions geared towards reducing symptoms of depression are likely to carry concurrent and prospective benefits for psychological functioning, namely the ability to be present in the moment and to savor.

## 1. Introduction

The study of mindfulness and mental health has flourished in psychological science over the recent decade [1,2,3,4]. According to Kabat-Zinn [5], mindfulness refers to the arising awareness of paying attention to the present moment without judgment and on purpose. In the mindfulness-to-meaning theory, Garland et al. [1] postulate that mindfulness is associated with psychological well-being and meaning in life through a spiral of processes, including reduced habitual and autopilot reactions to stress, greater cognitive reappraisal, and greater savoring of positive experiences. Recent studies have also suggested that trait mindfulness and mindfulness-based practices are associated with fewer symptoms of depression and anxiety [6,7,8,9,10,11]. Thus, the relations between mindfulness, psychological distress, and mental well-being have been well established in the literature.

While ample evidence suggests that mindfulness is beneficial to psychological health [6,7,8,9,10,11], a handful of studies have also shown that psychological health, as reflected by fewer depressive symptoms, is predictive of mindfulness [12,13,14]. Nevertheless, these findings appear to be inconsistent, especially on the bidirectional relation between mindfulness and depressive symptoms. In terms of significant findings, a study showed that depression severity was related to a lower level of mindfulness and that mindfulness mediated the longitudinal relation between the severity of depression and problematic smartphone use [12]. However, another study indicated that mindfulness and depressive symptoms did not predict one another from baseline to 3-month follow up but did show predictions of one another from 3-month to 6-month follow up [14]. In a third study [13], depressive symptoms consistently predicted mindfulness facets, including nonjudgment and nonreactivity to inner experiences, from baseline to 6-month follow up and from 6-month to 12-month follow up. Though depressive symptoms also predicted describing from baseline to 6-month follow up, they did not predict the rest of the facets, including observing and acting with awareness over time. As for the predictability of mindfulness on depressive symptoms, a significant effect was found for acting with awareness, but not other mindfulness facets, on depressive symptoms from 6-month to 12-month follow up (see also [15] for a meta-analytic review). As such, while depressive symptoms appear to be crucial for the worsening of some mindfulness facets, mindfulness facets are not always predictive of worsened depression in cross-lagged models. Given the mixed findings in the literature, the present study aims to disentangle the longitudinal relation between mindfulness and depressive symptoms among emerging adults.

Consistent with the mindfulness-to-meaning theory, recent findings have demonstrated that mindfulness is associated with a greater savoring of positive experiences [16,17,18]. Savoring refers to people’s propensity to be appreciative of the past (reminiscing), the present (savoring the moment), and the future (anticipation) [19,20]. Indeed, mindfulness is central to savoring, particularly to savoring the moment, as an awareness of the present is essential for the appreciation of it [21]. Not surprisingly, behavioral interventions have integrated components of mindfulness, savoring, and cognitive reappraisal techniques to reduce autopilot tendencies and foster behavior change [22,23,24,25,26].

While previous studies have demonstrated a direct association between mindfulness and savoring positive experiences, most of them are cross-sectional [3,27]. The studies that are longitudinal, however, did not examine the link between mindfulness and specific types of savoring, such as anticipation, savoring the moment, and reminiscing [18,28,29]. In terms of the directionality of effects, even though mindfulness and mindful decentering foster savoring [21,29], savoring the present moment may serve as a reward for people to be more mindful of the here and now. As such, this study aims to unravel the longitudinal relation between mindfulness and different types of savoring, including anticipation, savoring the moment, and reminiscing.

The association between savoring and psychological distress has also been well established in the literature [30,31,32], with a majority of the findings showing the prediction of savoring on lower distress, including depressive symptoms [3,33,34]. For instance, in a sample of older adults, a savoring intervention was found to reduce depression and negative affect and increase life satisfaction, subjective happiness, resilience, and positive affect [35]. Similarly, another study involving older adults showed that people who completed a savoring intervention with high fidelity had a greater reduction in depressive symptoms and greater improvements in happiness and resilience [36]. In a cross-sectional study involving undergraduate students, savoring, mindfulness, and self-compassion all mediated the relation between social support and depressive symptoms [3]. Interestingly, when the types of savoring were distinguished, only savoring the moment, not anticipation or reminiscing, mediated the cross-sectional relation between positive and negative affectivity and depressive symptoms among undergraduate students [37].

In terms of the opposite directionality of effects, a handful of studies have also suggested that people’s health, including fewer depressive symptoms and better-perceived health, were predictive of savoring [29,38,39]. However, the findings were somewhat mixed. For example, in a sample of parents of 8- to 12-year-old children, parental depressive symptoms longitudinally predicted lower relational savoring, i.e., parents’ ability to attend to, re-experience, recall, and process instances of positive connections with their child. Relational savoring was further associated with children’s greater physiological regulation [40]. As such, parental depression has direct implications for parents’ relational savoring as well as their child’s adjustment. In another study involving community-dwelling older adults, perceived health at baseline predicted anticipation and reminiscing but not savoring the moment 2.5 years later [38]. Hence, the predictive effects of health on savoring varied depending on the type of savoring. To further illuminate the directionality of effects between savoring and depressive symptoms, the present study tested the longitudinal relations between different types of savoring and depressive symptoms among emerging adults.

Situated in the developmental period of emerging adulthood, the present study examined cross-lagged relations between mindfulness, savoring, and depressive symptoms in a Chinese sample from Hong Kong. Emerging adulthood is a transition period in between adolescence and adulthood, characterized by identity exploration, instability, and possibilities [41,42]. Of note, people in this developmental period are prone to mental health problems including depression (e.g., [42,43]). As a result, identifying mental health correlates during this period is tremendously important. Based on the literature, we hypothesized that mindfulness would predict greater savoring, including reminiscing, savoring the moment, and anticipation, as well as fewer depressive symptoms. We also hypothesized that savoring would predict fewer depressive symptoms. Finally, we hypothesized that depressive symptoms would predict lower levels of mindfulness and savoring. However, we did not set an a priori hypothesis for the effects of savoring on mindfulness, given the lack of evidence in the literature.

## 2. Materials and Methods

### 2.1. Participants

One hundred and eighty Chinese university students of 18–27 years old (*M* = 21.12, *SD* = 2.02; 89.44% female) were recruited via online platforms and mass emailing. The retention rate was 92.78% (*n_T3_* = 167) from Time 1 (T1) to Time 2 (T2), with each time spanning three months. The present study was approved by the Human Research Ethics Committee of The Education University of Hong Kong. Informed consent was sought prior to the administration of the study. Upon completion of all time points, participants received HKD 200.00 supermarket coupons (~USD 25.64) as compensation.

### 2.2. Measures

#### 2.2.1. Mindfulness

The 10-item Cognitive and Affective Mindfulness Scale-Revised (CAMS-R; [44]) was used to assess mindfulness. Sample items included “It is easy for me to concentrate on what I am doing.” and “I am able to accept the thoughts and feelings I have.” Participants responded on a 4-point scale from 1 (*rarely*/*not at all*) to 4 (*almost always*). The items were averaged to form a composite score of mindfulness, with higher scores indicating a greater level of mindfulness. The CAMS-R was previously validated in a Chinese community sample [45]. Cronbach’s alpha = 0.86 at T1 and 0.89 at T2.

#### 2.2.2. Savoring

The 24-item Savoring Beliefs Inventory (SBI; [46]) was used to measure savoring positive experiences on a 5-point scale from 1 (*strongly disagree*) to 5 (*strongly agree*). The SBI has three subscales, including Anticipation, Savoring the Moment, and Reminiscing. The 8-item Anticipation subscale included sample items, such as “I feel a joy of anticipation when I think about upcoming good things” and “For me, anticipating what upcoming good events will be like is basically a waste of time.” The item scores were averaged to form a composite score of anticipation, with higher scores indicating greater anticipation. The 8-item Savoring the Moment subscale included sample items, such as “I know how to make the most of good time.” and “I feel fully able to appreciate good things that happen to me.” The item scores were averaged to form a composite score of savoring the moment, with higher scores indicating greater present focus. The 8-item Reminiscing subscale included sample items, such as “I can make myself feel good by remembering pleasant events from my past.” and “I like to store memories of fun times that I go through so that I can recall them later.” The item scores were averaged to form a composite score of reminiscing, with higher scores indicating greater reminiscing. The measures had been used previously with Chinese samples with adequate reliability [47]. Cronbach’s alpha = 0.77 at T1 and 0.81 at T2 for Anticipation, 0.81 at T1 and 0.83 at T2 for Savoring the Moment, and 0.83 at T1 and 0.80 at T2 for Reminiscing.

#### 2.2.3. Depressive Symptoms 

The 10-item Patient Health Questionnaire-9 (PHQ-9; [48]) was used to measure the severity of depression over the past two weeks on a 4-point scale from 0 (*not at all*) to 3 (*nearly every day*). Sample items included “Feeling down, depressed, or hopeless” and “Poor appetite or overeating”. The item scores were summed to form a composite score of depressive symptoms, with higher scores indicating higher severity. The measure has previously been validated in a Chinese community sample ([49]). The measure demonstrated adequate internal consistency with Cronbach’s alpha = 0.89 at T1 and 0.91 at T2.

#### 2.2.4. Sociodemographic Information 

Participants’ ages and genders were obtained from the questionnaire. Participants’ family income per month was measured on a 7-point scale (1 = 10,000 RMB or less; 2 = 10,001–20,000 RMB; 3 = 20,001–30,000 RMB; 4 = 30,001–40,000 RMB; 5 = 40,001–50,000 RMB; 6 = 50,001 RMB or above).

### 2.3. Data Analysis

Correlations, means, and standard deviations were examined in the preliminary analyses. A cross-lagged panel model using MPLUS, Version 8 [50], was conducted to investigate the changes in mindfulness, anticipation, savoring the moment, reminiscing, and depressive symptoms, respectively. The changes in variables were modeled by the stability coefficients between time-adjacent measures, e.g., T1 mindfulness was used to predict Time 2 (T2) mindfulness. Cross-lagged associations were modeled by predictions between the variables, e.g., T1 mindfulness was used to predict T2 depressive symptoms. Residual variance was allowed to covary to account for shared measurement variance. Sex, age, and monthly family income were included as covariates for all variables under study. Maximum likelihood method was used to investigate the model fit to the observed matrices of variance and covariance. Full information maximum likelihood estimation was used to handle missing data, given that the data were missing completely at random (MCAR), as indicated by Little’s MCAR test: χ^2^ (2495) = 2595.22, *p* = 0.08.

## 3. Results

Table 1 shows the zero-order correlations, means, and standard deviations. Specifically, the core variables under study were significantly associated with one another over time, with *p*s < 0.05. As for the demographic variables, being female was associated with greater mindfulness at T1 and T2 (*r* = 0.19, *p* = 0.01 and *r* = 0.17, *p* = 0.03, respectively). Being younger was associated with lower anticipation at T1 and T2 (*r* = −23, *p* = 0.002 and *r* = −0.22, *p* = 0.01, respectively). Finally, greater family income was associated with greater mindfulness at T1 (*r* = 0.19, *p* = 0.04).

In testing the cross-lagged analysis, the path model fit adequately to the data, with χ^2^(8) = 8.32, *p* = 0.40, CFI = 1.00, TLI = 1.00, RMSEA = 0.01; SRMR = 0.03 (see Figure 1). The core findings indicated that T1 savoring the moment positively predicted T2 mindfulness (β = 0.25, *p* = 0.02). T1 depressive symptoms negatively predicted T2 mindfulness (β = −0.21, *p* = 0.01) and T2 savoring the moment (β = −0.15, *p* = 0.03). The autoregressive control variables of mindfulness, anticipation, savoring the moment, reminiscing, and depressive symptoms were significant, with *p*s < 0.001 (see Table 2 for details).

## 4. Discussion

The present study broadens the literature by disentangling the effects between mindfulness, savoring, and depressive symptoms among emerging adults. Importantly, the longitudinal findings lend support to the significance of savoring the moment in enhancing mindfulness. They also indicate that depressive symptoms are crucial for diminishing mindfulness and savoring the moment. Contrary to previous research indicating that mindfulness and savoring are both beneficial to psychological functioning [1,23,24,25], the cross-lagged analysis revealed, instead, an opposite directionality of effect, after controlling for age, gender, and family income. Consequently, the findings brought forth unique implications for prevention and intervention in emerging adulthood. 

Somewhat surprisingly, mindfulness did not predict different types of savoring. The null findings contradicted the mindfulness-to-meaning theory [1] and existing studies that show a positive effect of mindfulness on savoring [18,51]. Conversely, savoring the present moment, but not anticipation or reminiscing, was predictive of greater mindfulness. While this finding is new to the literature, it is consistent with our speculation that savoring the present moment can serve as a reward for people to be more mindful of the here and now. To add specificity to the relation between mindfulness and savoring, or the lack thereof, researchers should recruit a larger sample to conduct cross-lagged analyses between different types of savoring and mindfulness facets, including nonjudgment and nonreactivity to inner experiences and describing, observing, and acting with awareness. 

Unique to the present findings is that depressive symptoms were predictive of mindfulness and savoring the moment, over and above autoregressive effects, age, gender, and family income. These findings partially agree with previous research that showed that depressive symptoms were predictive of some but not all facets of mindfulness among adolescents [13]. They also corroborate the mindfulness-to-meaning theory [1], which suggests that correlates of depression such as cognitive vulnerabilities and negative stress appraisal [52,53] reduce the state of mindfulness. Future research is, again, necessary to examine the longitudinal associations between depression and mindfulness facets across developmental periods. Importantly, identifying why, when, and how depression is predictive of mindfulness can inform existing theories of mindfulness and mental health [1]. As for savoring, the present findings contradicted another study involving community-dwelling older adults, which showed that perceived health at baseline predicted anticipation and reminiscing but not savoring the moment 2.5 years later [38]. Given that the present research examined depressive symptoms as a predictor (instead of general health) during emerging adulthood (instead of older adulthood) at a 3-month time lag (instead of a 2.5-year lag), the discrepancies between the findings might have been due to a range of variables. Hence, further studies are necessary to identify the moderators between health and savoring. Based on the present findings, the major implication is that, upon reductions in depression severity, people are likely to also experience a greater awareness of paying attention to and appreciating the present moment without judgment and on purpose. In addition, this study informs clinicians and practitioners of the importance of lowering the severity of depression such that people can have a greater capacity for present-moment awareness and appreciation.

In contrast to previous studies [2,3,4,37], neither mindfulness nor savoring predicted depressive symptoms in the cross-lagged model. The null findings were surprising, as they contradicted intervention studies showing the benefits of mindfulness [54,55,56] and savoring [35,36] in reducing depressive symptoms. Given that depressive symptoms are stable over a three-month period [57], the null findings might have been due to the stability of depression over a short period of time. Hence, future studies are necessary to replicate the present study by increasing the lag between time points. 

The present findings must be interpreted in light of several limitations that point to future research directions. First of all, the present study included self-report measures to assess the variables of interest. Some of the variables, such as the ability to be present in the moment and to savor, should not be measured by a questionnaire alone due to self-report bias. To reduce self-report bias, future research should employ a multi-method, multi-reporter approach. Second, the majority of the participants were female (89.44%). As such, researchers should collect a more gender-balanced sample to further investigate the gender effect. Third, the time lag was short between assessment points at three months apart. Importantly, we might have underestimated the stability over the short time lag, potentially leading to attenuated relations between different variables. To examine the long-term effects among these variables, researchers should consider increasing the lag to 6 or 12 months. They should also collect additional data for multiple time points to examine the trajectories of mindfulness, savoring, and depressive symptoms. Fourth, an a priori power analysis should have been conducted before the start of the study to justify the sample size. 

Ample theories and evidence to date have suggested that interventions involving mindfulness and savoring are beneficial to psychological functioning [1,23,24,25,33,35,54,55,58]. Through cross-lagged analysis, the present study showed that a lower level of depression can also foster greater mindfulness and savoring the moment, particularly among emerging adults in the Chinese context. This study informs clinicians and practitioners of the implications of reducing the severity of depression. Prevention and intervention studies merit future investigations. 

## Figures and Tables

**Figure 1 ijerph-20-05568-f001:**
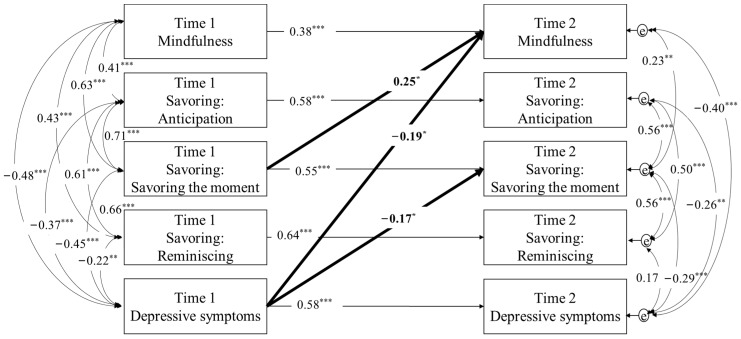
Path model between mindfulness and savoring. χ^2^(8) = 8.32, *p* = 0.40, CFI = 1.00, TLI = 1.00, RMSEA = 0.01; SRMR = 0.03. Age, gender, and family income were analyzed as covariates but are not included in the Figure for clarity. Non-significant paths are depicted in dashed arrows for clarity. * *p* < 0.05, ** *p* < 0.01, *** *p* < 0.001.

**Table 1 ijerph-20-05568-t001:** Zero-order correlations, means, and standard deviations of the variables under study (N = 180).

Variable	1.	2.	3.	4.	5.	6.	7.	8.	9.	10.	11.	12.	*M*	*SD*
1. Gender (0 = male; 1 = female)	-												-	-
2. Age	−0.01	-											21.08	2.02
3. Family Income	0.12	−0.06	-										-	-
4. T1 Mindfulness	0.19 *	0.02	0.18 *	-									2.80	0.48
5. T1 Anticipation	−0.03	−0.23 **	0.16	0.40 ***	-								3.21	0.51
6. T1 Savoring the Moment	0.05	−0.10	0.15	0.63 ***	0.71 ***	-							3.32	0.57
7. T1 Reminiscing	0.12	−0.12	0.09	0.43 ***	0.61 ***	0.66 ***	-						3.45	0.53
8. T1 Depressive Symptoms	−0.05	0.13	−0.07	−0.48 ***	−0.37 ***	−0.45 ***	−0.22 **	-					6.18	5.00
9. T2 Mindfulness	0.17 *	−0.13	0.16	0.57 ***	0.30 ***	0.47 ***	0.31 ***	−0.43 ***	-				2.79	0.51
10. T2 Anticipation	0.02	−0.22 **	0.15	0.32 ***	0.64 ***	0.53 ***	0.43 ***	−0.32 ***	0.29 ***	-			3.24	0.53
11. T2 Savoring the Moment	0.06	−0.06	0.07	0.50 ***	0.53 ***	0.69 ***	0.49 ***	−0.42 ***	0.50 ***	0.69 ***	-		3.30	0.54
12. T2 Reminiscing	0.09	−0.49	0.08	0.35 ***	0.43 ***	0.48 ***	0.70 ***	−0.19 *	0.31 ***	0.57 ***	0.65 ***	-	3.41	0.54
13. T2 Depressive Symptoms	−0.16 *	0.11	−0.12	−0.45 ***	−0.31 ***	−0.42 ***	−0.20 *	0.67 ***	−0.60 ***	−0.38 ***	−0.50 ***	−0.26 **	6.61	5.53

Note. * *p* ≤ 0.05, ** *p* ≤ 0.01, *** *p* ≤ 0.001.

**Table 2 ijerph-20-05568-t002:** Parameter estimates of the hypothesized pathways.

Parameter	Unstandardized *B* (*SE*)	Standardized β
T1 Mindfulness		
→ T2 Savoring: Anticipation	0.03 (0.08)	0.03
→ T2 Savoring: Savoring the Moment	0.12 (0.09)	0.11
→ T2 Savoring: Reminiscing	0.05 (0.08)	0.05
→ T2 Depressive Symptoms	−1.14 (0.93)	−0.10
T1 Savoring: Anticipation		
→ T2 Mindfulness	−0.17 (0.10)	−0.17
→ T2 Depressive Symptoms	0.78 (0.92)	0.07
T1 Savoring: Savoring the Moment		
→ T2 Mindfulness	0.23 (0.10) *	0.25 *
→ T2 Depressive Symptoms	−1.52 (0.96)	−0.15
T1 Savoring: Reminiscing		
→ T2 Mindfulness	0.02 (0.08)	0.02
→ T2 Depressive Symptoms	0.69 (0.81)	0.07
T1 Depressive Symptoms		
→ T2 Mindfulness	−0.02 (0.01) **	−0.21 **
→ T2 Savoring: Anticipation	−0.01 (0.01)	−0.06
→ T2 Savoring: Savoring the Moment	−0.02 (0.01) *	−0.15 *
→ T2 Savoring: Reminiscing	−0.01 (0.01)	−0.06
Sex (0 = male; 1 = female)		
→ T2 Mindfulness	0.14 (0.11)	0.08
→ T2 Savoring: Anticipation	0.03 (0.11)	0.02
→ T2 Savoring: Savoring the Moment	0.04 (0.10)	0.02
→ T2 Savoring: Reminiscing	0.03 (0.10)	0.01
→ T2 Depressive Symptoms	−2.27 (1.06) *	−0.12 *
Age		
→ T2 Mindfulness	−0.02 (0.02)	−0.09
→ T2 Savoring: Anticipation	−0.03 (0.02)	−0.10
→ T2 Savoring: Savoring the Moment	0.01 (0.02)	0.02
→ T2 Savoring: Reminiscing	0.00 (0.02)	0.02
→ T2 Depressive Symptoms	0.08 (0.16)	0.03
Family Income		
→ T2 Mindfulness	0.02 (0.03)	0.05
→ T2 Savoring: Anticipation	0.02 (0.03)	0.04
→ T2 Savoring: Savoring the Moment	−0.02 (0.03)	−0.04
→ T2 Savoring: Reminiscing	0.01 (0.03)	0.02
→ T2 Depressive Symptoms	−0.17 (0.31)	−0.04
Autoregressive Control		
T1 → T2 Mindfulness	0.39 (0.09) ***	0.36 ***
T1 → T2 Savoring: Anticipation	0.59 (0.06) ***	0.57 ***
T1 → T2 Savoring: Savoring the Moment	0.52 (0.06) ***	0.55 ***
T1 → T2 Savoring: Reminiscing	0.64 (0.05) ***	0.64 ***
T1 → T2 Depressive Symptoms	0.68 (0.08) ***	0.60 ***

*Note*. * *p* =/< 0.05, ** *p* =/< 0.01, *** *p* =/< 0.001.

## Data Availability

The data presented in this study are available on request from the corresponding author. The data are not publicly available due to ethical constraints.

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
