# Peer review of "Disentangling the Effects of Mindfulness, Savoring, and Depressive Symptoms among Emerging Adults"

_ijerph, 2023, doi:10.3390/ijerph20085568_

Round 1

Reviewer 1 Report

First of all, thank you very much for giving me the opportunity to review the manuscript titled: Untangling the Effects of Mindfulness, Taste, and Depressive Symptoms Among Emerging Adults.  

 It is a very interesting job.    

Their introduction is very well structured.  In addition, it provides very relevant and updated information on the variables under study.  The only suggestion I have regarding this section is to remove the subtitles.  I think it would be interesting to write the introduction without them.   

In relation to the methodology, it would be convenient to include in the instruments a section of sociodemographic variables that include age, gender and family income, since they are later included in the model as covariates.  add information about variable household income.    

Regarding the results, in Table 1 it would be convenient to add the mean and the standard deviation as a column and not as a row.  This would allow us to eliminate column 13. You put mean and standard deviation in gender and household income.  I don't understand when they are categorical variables how they have means.  Is it normal for women?  Man?  I would change it to n and %.    

The discussion is complete and correct.

Reviewer 2 Report

While this is is a great paper, well grounded in theory, a sophisticated methodological approach and with a high practical relevance, I require the following changes:

1) with the literature review (relating to the intro and discussion) please check for updates as I did not see any papers from the year 2023.

2) how was missing data from the study participants handled who dropped out after the first measurement point? I miss a dropout analysis and an appropriate data imputation method (FIML would need more justification and I fear a different method is needed here).

3) the theoretical considerations need to be integrated into the discussion as well and discussed accordingly.

4) Please torn down your conclusions and implications to the actual empirical findings: Suggestions re interventions is fine, but relating to the ability to be present in the moment and to savor is more than what can be measured with a questionnaire. And the sentence "Taken together, reducing depressive symptoms may be crucial in fostering psychological well-being." is too generic. Please alter the text accordingly.

Reviewer 3 Report

Thank you for the opportunity to review this manuscript.

The abstract is clear and informative. The introduction outlines the relevant literature while elucidating gaps in knowledge that the present study will fill. The method was clearly written and included all the necessary information needed to attempt to replicate the authors’ findings. The results were clearly presented. The tables and the figure were helpful. The discussion was engaging and clear. I have only two suggestions for edits:

Throughout the manuscript, please double-check there are both opening and closing brackets surrounding citations (e.g., p. 3, line 113). It also appears there are times that the authors state something like: According to [#], where they only insert the citation number and do not provide the name of the cited scholar (e.g., the second sentence of the introduction). I realize that the in-text citations will be bracketed links to the reference page and each of them do not need the cited person identified, however, I think stylistically it would still be useful in sentences with these “According to…” structures that the scholar being cited is identified by name, e.g., According to Kabat-Zinn [#], ….

p. 7, line 209: “preventions” should be changed to “prevention”

Reviewer 4 Report

Dear authors

Thank you very much for your manuscript. The manuscript meets the requirements of the journal and the topic is relevant. The introduction is comprehensive and introduces the topic well. In the description of the studies, I do not understand why you did not formulate an a priori hypothesis. The justification that evidence for this is lacking seems inconclusive to me. With regard to materials and methods, you describe that you interviewed 180 students. Why is there no power calculation for the study? The presentation of the scales used is comprehensive and comprehensible. The description of the data analysis is complete. I find the presentation of the results (lines 181 to 199) mainly on the basis of tables 1 and 2 and figure 1 too brief. The presentation of zero-order correlations, means and standard deviations in Table 1 without a description of the participants seems unusual to me. The discussion is comprehensible. However, with regard to the presentation of the limitations, the very short observation period of three months and the limitations of the cross-lagged analysis method should be addressed.
